# Phenomenology of Quranic Corporeality and Affect: A Concrete Sense of Being Muslim in the World

**Valerie Gonzalez** 

Independent Researcher, London SW5 0QF, UK; valerie.gonzalez152@googlemail.com

**Abstract:** It is a matter to ponder that, among the three Abrahamic monotheisms, Islam places the greatest ontotheological distance between the human and the divine. While God is the ground of being Muslim, Islam excludes theophany and prohibits any tangible association between the divine and anything in the material world. God's mode of manifesting Himself to His creatures has consisted of the most fleeting and discorporate of all means of communication, namely, sound. His words gathered in the Qur'an thus form a non-solid verbal bridge crossing over that unfathomable distance. One could then think that the relationship between the unique Creator and His creatures relies only on the strength of a blind faith founded on a dry, discursive pact. Arguing his "idea of an anthropology of Islam", Talal Asad did posit that this religion and its culture form "a discursive tradition". Exclusively focused on the mental modes of knowledge acquisition, this cognitivist verbalist characterization has become a certitude in Islamic studies at large. Yet, it is only a half-truth, for it overlooks the emphatic involvement, in the definition of this tradition of Islam, of the non-linguistic phenomenality of experience that implicates the pre-logical non-cognitive double agency of affect and sensation in the pursuit of divine knowledge. This article expounds this phenomenology of the Qur'an in using an innovative combination of philosophical and literary conceptualities, and in addressing some hermeneutical problems posed by the established Quranic studies.

**Keywords:** Islam; Quranic studies; religious phenomenology; sacred aesthetics; hermeneutics; epistemology



## 1. Introduction: Addressing a Half-Truth

It is a matter to ponder that, among the three Abrahamic monotheisms, Islam places the greatest ontotheological distance between the human and the divine. While God is the ground of being Muslim, Islam excludes theophany and prohibits any tangible association between the divine and anything in the material world. In the Islamic tradition of divine revelation, God's mode of manifesting Himself to human beings has consisted of the most fleeting and discorporate of all means of communication, namely, sound.[1] His words gathered in the Qur'an thus form a non-solid verbal bridge crossing over that unfathomable distance. One could then think that the relationship between the unique Creator and His creatures relies only on the strength of a blind faith founded on a dry, discursive pact. Famously arguing his "idea of an anthropology of Islam", Talal Asad did posit that this religion and its culture form "a discursive tradition".[2] Exclusively focused on the mental modes of knowledge acquisition, this cognitivist verbalist characterization has become a certitude in Islamic studies at large. Yet, it is only a half-truth, for the following reasons:

Above all, albeit in different ways, Judaism and Christianity equally form a discursive tradition upon which Islam has built its own, in its own unique terms of autoferentiality, as Anne-Sylvie Boisliveau explains in *Le Coran par lui-même*.[3] Therefore discursivity is not a particularity of Islam. Then, if Asad's positing nevertheless makes sense in the light of both this argumentative feature of self-definition and Islam's dogmatic logocentrism, it overlooks a major and unparalleled ipseitic dimension of "the Islamic tradition"—this designation broadly signifying the Muslim mode of being in the world based on a built relationship

between the divine and the human.[4] This dimension is the emphatic involvement in this tradition of the non-linguistic phenomenality of experience, which implicates the pre-logical non-cognitive double agency of affect and sensation in the pursuit of divine knowledge.[5] To get a sense of this incompleteness of Asad's predicate, suffice it to recall the crucial role in Islamic theology, *falsafa*, and Sufi literature of the polyvalent somatic concepts of "the heart", "love", and "beauty", signaling Islam's both cognitive and affective requirements from the faithful.[6] In a previous work, I have named this aspect of the condition of being Muslim "affective understanding" (Gonzalez 2020, p. 2). It is important to note, however, that this mode of understanding expected from the believer specifically concerns the experiential aspect of the divine–human relationship; therefore, it is to be distinguished from the religious ethics of love common to the three Abrahamic creeds.

Manifesting itself powerfully in the Qur'an, this mute affective–sensorial component of the Islamic tradition has yet to be fully grasped as the equal counterpart to its discursive dimension. Although the Quranic message is, by definition, a construct of language and, a fortiori, a discursive product meant to induce noetic processes of reception, it indeed equally operates in the pre- or para-conscious phenomenality of "the felt"—the concept of the felt consisting of the rich compound of bodily and psychic feelings and emotions termed *le senti* and *le ressenti*, respectively, in French phenomenology.[7] More than that, stimulating the subject's affectivity by unprecedented literary means, the Qur'an sets itself apart from the other Abrahamic scriptures on the subject of the phenomenal existence. This essay aims to uncover these unidentified literary means, allowing Islam's holy book to function as much in the instructive field of narration as in the empirical field of the lived experience. To that effect, attention is paid to the Qur'an's mechanisms of verbal–sonic communication and its phenomenology of the sacred located in both the text's literal semantics and what I call "its texture". Poetry being the mode of affective expression by excellence in the verbal domain, a critical reflection on its Quranic use is central to this endeavor.

## 2. Definition of the Quranic Texture

The Quranic texture constitutes the unobvious stylistic layer of the holy book that, imbued with certain phenomenological properties, specifically stimulates the faithful's sensory and affective faculties. It thus turns the reception of God's words into a fully lived "total experience".[8] This empirical functioning is an integral part of the literary mechanisms that the Islamic divine message uses to set work for the addressees' spiritual awakening and learning process. Premised upon this assertion, I argue that, through the activation of the entire spectrum of the human perceptive capacities—intellective, perceptual, emotional, and psychical—the Qur'an allows for a profoundly humanistic intimacy with God in an otherwise ungraspable, disembodied, and abysmally distanced divine–human ontotheological relationship. But of what exactly does this Quranic texture consist?

Evoking the sensuous quality of hapticity, the term "texture" designates a certain set of phenomenological agents in and of the Quranic text that work together, albeit beyond the narrative's literality. This texture corresponds, for instance, to what Sebastian Günther and Todd Lawson intuitively and variably term in *Roads to Paradise* the "powerful ways", "unique way", or "evocative", "explicit", and "vivid" manner with which the Qur'an delivers its message. For example, they write "The hope is to demonstrate that these topics and literary functions are among those parts of the Quran that carry the apocalyptic theme most vividly" … "The Quran speaks of death and resurrection, of the end of this world, and of the world to come more than any other major scripture, and it does so in a remarkably explicit and evocative manner" (Günther 2017, p. 181). In sum, the duality of the Quranic texture versus the Quranic text proceeds from the essential distinction between meaning and expression and between logos and pathos, as the texture concept refers not to the text's apparent content itself, but to the mode of communication and presentation by means of which this content constitutes itself in the beholder's consciousness.[9] Still, as we are dealing with the verbal, the Quranic texture is necessarily located within its multilayered semantics, at a deeper level than the text's literality.

To clarify this conceptuality with an analogy, the Quranic texture is to the text's verbal world what the compound of style and pictorial metaphysics is to the visual world of figurative painting. This compound serves to present the painting's subject matter in a certain way, be it a Madonna, a portrait, a landscape, or a still life; therefore, it is not to be confused with this subject matter itself. While style is the product of the manipulation of visual appearances (thanks to instruments such as palettes, brush strokes, contouring lines, perspective, flatness, etc.), pictorial metaphysics refers to the conception of the worldview projected through the representation/subject matter, which may encompass realism, naturalism, expressionism, idealism, and symbolism. To summarize, the distinction between meaning and expression or logos and pathos in textuality equates to the difference between representation/subject matter and stylistic–metaphysical presentation in figurative painting. As the manner of apprehension of the text or picture's subject matter by the beholder depends entirely on the modality of expression/presentation through which this subject matter offers itself up to consumption, it is not possible to fully understand this text or picture without paying attention to this modality.

Furthermore, like the Quranic mode of utterance with which it forms a twin phenomenology of communication, the Quranic texture is instrumental to both the cognitive efficiency and psychology of reception of the text. Bearing this in mind, the double question of the mode of reception the Qur'an aims to produce and the tools it employs for this purpose lies in the core of the problematic of the Qur'an as a phenomenological object of affective–sensory experience with a texture. To define more precisely this type of object in the non-corporeal domain of the verb, it possesses "expressive" qualities allowing it, like material art, to direct perception in the region of emotions, feelings, and sensations.[10] Tackling this problematic, however, requires a thinking and working method that does not pertain to the Quranic studies' established pragmatics and objectives. While philology, linguistics, and traditional hermeneutics are of essential use to this scholarship focused on the examination of the text's literary history, literal semantics, and structure of language, these scholarly practices become insufficient when it comes to penetrating the phenomenological levels of Quranic expressiveness. As if this situation was not already challenging enough, a ready-made method that would help with this type of task simply does not exist—it must be invented. Echoing Roland Barthes who, in *Camera Lucida*, took issue with the ways in which photography was being talked about in the aesthetic criticism of the 1970s, we could ask "why mightn't there be, somehow, a new science for each object?" (Barthes 2000, p. 8).

### 3. Studying the Quranic Phenomenology of Sensation and Affect: A Methodological Challenge

While inventing a working method necessarily implies a risky unorthodox challenging of the established epistemic habits, it also allows for a free exploration of intuitions and a choice of critical tools based on a bold stance vis-a-vis the various conventional approaches to the topic observed. This unusual epistemic positioning defines this and other works of mine as, to quote Barthes again, I have been engaged in "a desperate resistance to any reductive system" (Ibid, p. 8). Thus, to address the Quranic problem presented above, I will use the cross-field methodology that I have been developing and implementing over the years. This methodology follows three main epistemic axes in intersection with one another: the study of Islamic art and visuality; the study of the Islamic faith and the Qur'an; and the critical, aesthetic, and philosophical inquiry in general, especially phenomenology that is concerned with the structures of consciousness, intentionality, and perception.[11] However, beyond the impetus of scholarly pragmatics, this choice of combined specialisms has been motivated by a deeper consideration about Islamic visual culture. As I see it, this culture is the product of both Islamic metaphysics and a profound level of religious affect shaping the Muslim life condition in all of its aspects. It consequently reflects and expresses the Islamic faith, even in the case of objects, architecture, and designs with no apparent devotional function. Islamic visuality and religion thus being, in my view, indivisible, studying the one could not exempt itself from consulting the other. It is through this understanding of

the Muslim order of things that I came to form the intuition that not only does the Qur'an provide the primary conceptual source material for Islamic artistic creation, but also that it has itself an aesthetic ontological status exceeding the definition traditionally given to it as an inimitable linguistic masterpiece.[12] I intuited that it constitutes an artwork through and through: an artwork of divine order from the viewpoint of the Islamic phenomenology of the sacred. However, this view positing religion as an integral part of art in Islam, and vice versa, is far from being consensual. It comes up against an important epistemic problem on the subject of the sacred in Islamic cultural studies that one cannot ignore, as it pertains to this essay's conceptual background (see Ahmed 2015).

A deep dichotomy, whose history goes back to the mid-20th century, indeed affects said studies with, on the one side, the scholarship harboring this view that I endorse and, on the other side, the proponents of a divergent conception of Islamic culture separating the religious practice and its devotional material supports from the rest of the cultural life and products. I call this positioning "secularist" because it divides the traditional Islamic cultural space according to the binary of the religious versus the secular that, as we know, originates from post-Enlightenment rationalist thought. Given that the latter has shaped the hermeneutical practices in the humanities at large, it is no surprise that this secularist conception's proponents form the mainstream scholarship and curatorship, with those refuting this idea of secularity's existence in this space creating a countercurrent struggling against this dominance. Evidence of this situation can be found both in the studies considered to be the most authoritative and in the displays of Islamic art in world-class museums functioning as global museological paradigms, such as the Metropolitan Museum of Art, the Louvre, and the British Museum. Both conveyors of knowledge on Islam characteristically pass over or stifle the sacred in the narrative on the non-religious cultural material.[13] For example, in the Victoria and Albert Museum in London, a textbox labels "Mameluk Secular Art" an installation of a priori non-cultic objects made under the Mameluks' patronage. But whether or not there exists such a thing as "Mameluk secular art", it is no accident that this label appears on the walls of a European museum. Such a labeling only signals the highly significative configuration of the epistemic cleavage in question, whereby the mainstream scholarship reigns over the most prestigious art and learning institutions in the West, and the countercurrent is in great part active in the Muslim world.[14] It should be noted, though, that the Western secularists oppose the scholars hailing from the Muslim world by essentially ignoring their work, as if it had no weight on the matter. The glaring paucity of references to these scholars in this Western scholarship's findings attests to this attitude.

This epistemic cleavage, which also contributes to the problematic institutional disconnection between Islamic religious studies and the history of Islamic art and visuality, is not the only challenge to consider in the enterprise of turning the intuition of the Qur'an as a work of art into a solid hermeneutical proposition. The question also poses itself of gathering state-of-the art critical equipment that may satisfactorily support the analyses of both the religious and aesthetic–artistic matters involved in this Quranic problem. It is at this point, then, that I resort to the third and most unconventional of my "toolboxes", to use a famous Deleuzian term—namely, the one that freely draws from a whole range of aesthetic theories, criticisms, philosophies, and literatures. Mostly Western and modern, this etic theoretical material constitutes an invaluable reservoir of useful conceptualities and definitions of aesthetic processes and phenomena that cannot be found in the emic primary sources, nor in the conventional discourses on Islamic culture, art, and religion.[15] However, this methodological approach comes up against two major difficulties due to its transculturality:

The first difficulty is that the incursion in the Western modern philosophical–critical domain necessitates a tricky navigation among a wealth of complex trends of thought and aesthetic movements that are not always "purely theoretical", in the sense of being free of cultural determinants. By necessity, then, the selection of source material made of

such determinants must rely on this material's demonstrated ability to be transculturally transferable and applicable to Muslim culture.

The second difficulty has to do with certain circles of the scholarly establishment's reserve or disagreement concerning the use of etic Western materials for the investigation of an Islamic topic. To experts in the history of Islamic art and visual culture—the field to which I primarily belong—mixing etic and emic sources is highly contentious, especially in the current general context of decoloniality. Given this etic material's Western modern origin, the charge of anachronism, neo-Orientalism, and cultural inappropriateness invariably comes up. I have addressed this issue theoretically many times so that, here, in response to this potential objection, I can simply let the reader evaluate the results of this inquiry on the Qur'an's phenomenological complexity.

## 4. Arousalism of the Qur'an as Text–Artwork

As sophisticated as it is linguistically, the Qur'an is more than a text saying things. Analogous to a solid material artwork, it operates on the perceptive–perceptual terrain by means of both its texture and its sonicity as an orally performed scripture. This textural and sonic expressivity produces an affective–sensory stimulation of remarkable intensity— a procedure called "arousalism" in aesthetic theory.[16] In Western thought, this concept designates that type of stimulation in the aesthetic workings of any object, natural or cultural, in any empirical circumstance. In art, arousalism is more specifically associated with the key moment of the Western artistic–cultural history of "modernism" that began in the late 19th century and fully unfolded in the early 20th century. Realizing the potency of the forms' non-discursive affective and perceptual properties in art-making, some modernist artists endeavored to jettison naturalist figurative representation as the classical paradigm of Western plastic expression. They thus embarked on the path of abstraction. One of the pioneers of this aesthetic movement in the West was the famous painter and theorist Wassily Kandinsky who, in 1912, wrote a landmark aesthetic treatise, *Concerning the Spiritual in Art*, in which he theorized the principles subtending arousalism as per this excerpt on color:

> "Colour directly influences the soul. Colour is the keyboard, the eyes the hammers, the soul is the piano with many strings. The artist is the hand that plays, touching one key or another purposively, to cause vibrations in the soul" (Kandinsky 1977, p. 45) quoted by (Benenti 2020, p. 31).

To transculturally broaden this framework of arousalism, "rasa" is an arousalist concept in ancient Indic-Indian aesthetics that signifies "a kind of contemplative abstraction in which the inwardness of human feelings suffuses the surrounding world of embodied forms".[17] It applies to theatre, poetry, and all arts. Observed through the lens of these theories, the Qur'an itself involves the greatest artist and producer of rasas of all: God Himself. As Sebastian Günther observes in borrowing Andrew Rippin's wording, "the multiplicity of colors, hues, and shades is 'evidence for God's handiwork in creation'." (Günther 2017, p. 206). Similarly, as a divinely created object, the Quranic text has the capacity to generate sensations, affects, and rasas thanks to a rich array of arousalist tools, beginning with the one that is most immediately effective, namely, sound.

Seizing upon the body with the greatest directness, sound produces a uniquely immersive "encounter wherein a co-constitution of artwork and subject takes place", to paraphrase the contemporary sonic artist and theorist Will Schrimshaw (Schrimshaw 2017, p. 3). A powerful object of sonorous immersion when performed orally, the Qur'an induces a sonic arousalist phenomenology that has no parallel within the Abrahamic ritualistic traditions of sound. To briefly compare the practice of reading aloud the holy scripture in these traditions, while all share the unsurpassable Davidian musical paradigm, have roots in the ancient Middle East's oral liturgical customs, and are equally built upon this heritage, Islam became unique for the elimination of singing during the canonical rituals.[18] It channeled the arousalist power of sound quasi-exclusively toward the utterance of the holy scripture, alongside the equally unique thunderous call to prayer. Moreover, if what links the oral

delivery of the Qur'an, Torah, and New Testament is a controlled manner of pronouncing the text's words according to certain sonic parameters—be it the Hebrew cantillation, the Christian chanting and dramatic eloquence, or the melodic Arabic intonations—the Quranic utterance became distinct by unprecedentedly empowering, at another more elemental level beyond word pronunciation and without involving singing, the voice's timbre and pure sonorous effects.[19] Illustrating this hyper-valorization of the quality of the reciter's vocal organ and sonic skills, one hadith reports that the Prophet said to Abu Musa "O Abu Musa, you have been given a voice for recitation from the instruments of the house of David" (Ṣaḥīḥ Muslim, hadith. 793, cited by Badawi 2020, p. 139). In another hadith, Abu Musa reported, "I heard the Messenger of Allah saying, 'Allah does not listen so attentively to anything as He listens to the recitation of the Qur'an by a Prophet who recites well with a melodious and audible voice" (The Book of Virtues 8 2023, hadith 14).

This empowerment of the voice and sonic phenomenology imbues the Quranic oral reading with an ultra-penetrative sonicity, lending the auditory experience a numinous visceralness that the flow of the memorized recitation—this reading's ultimate form—renders most gripping and effective in soliciting the divine message's affective understanding. The spirit and the sense, if not the text's semantics itself, literally pass through this visceral sonic flux.[20] To once again put things in a transcultural perspective, in essence, the Quranic sonic arousalism converges with another arousalist principle traditionally attributed to the Indian monk Bodhidharma in the Chan and Zen Buddhist philosophical–aesthetic context. According to tradition, Bodhidharma strived for "a special transmission outside the scriptures; no dependence upon words and letters; pointing directly to the heart (intuitive mind) of man" (Quoted by Sirén 1963, p. 93)[21]. Although the Quranic performance is inextricably tied to the Quranic wording, following an analogous conception of affective spiritual communication, it deploys—alongside its sensational sonicity and within the textual field itself—other arousalist devices enabling "a special transmission" that points "directly to the heart". The verbal instrument of arousalism par excellence being poetry, the next discussion examines the Quranic texture's poetic component—more specifically, the symbolic genre revolving around the concept of the metaphor to which the scholarship refers recurrently (albeit, in my view, not always suitably).[22]

### 5. The Question of Symbolic Poetry in the Qur'an

The question of symbolization in the Qur'an is of utter importance for defining its arousalist system, as upon this definition depends the understanding of both the corporeality-based Quranic phenomenology of the sacred and what this very phenomenology might unravel regarding the Qur'an's ontology. Thus, in varied ways, all three members of the Abrahamic scriptural lore use poetry to spiritual ends—especially the symbolic genre, including metaphors, allegories, and parables that translate in words sensorial impressions, eloquently reinforce ideas, and eidetically express abstract thoughts in appealing concrete terms. Solomon's *Song of Songs* is the quintessence of that. Regarding the Qur'an, it is replete with metaphorical constructs, as in Q, Ibrahim, 14: 24:

> "Have you not considered how Allah presents an example, [making] a good word like a good tree, whose root is firmly fixed and its branches [high] in the sky?".[23]

Yet, once again, the Qur'an stands out for presenting a challenging symbolic structure. It is by no means self-evident that what, from a scholarly historical literary viewpoint, may appear as symbolic poetry systematically belongs to this genre in Islam's scripture from the ontotheological viewpoint. In the absence of the typical conjunction of the object symbolized and the metaphor itself through prepositions such as "like" and "as if", the question arises about the exact nature of some Quranic excerpts eloquently formulated in the form of colorful imageries, as in Q Ar-Ra'd 13: 35:

> "The example of Paradise, which the righteous have been promised, is [that] beneath it rivers flow. Its fruit is lasting, and its shade. That is the consequence for the righteous, and the consequence for the disbelievers is the Fire."

Accustomed as we are to what Aldous Huxley nicely calls "the cozy world of symbols", we might easily mistake some of these imageries for rhetorical devices functioning in the mode of the figure of speech (Huxley 2004, p. 33). Instead, these imageries may poetically (albeit not symbolically) describe real things, facts, and situations in the Islamic sacrality. Given the historical imprint of the Western mode of thinking on the scholarship on global cultures, it is indeed very possible that "our linguistic habits lead us into error", to cite Huxley again, and that we mischaracterize asymbolic Quranic verses as metaphorical or allegorical because "we are forever attempting to convert things into signs for the more intelligible abstractions of our own invention. But in doing so, we rob these things of a great deal of their native thinghood" (Huxley 2004, p. 61)[24].

The problem more particularly affects the study of the hyper-imagistic Quranic eschatological narrative and of the para-Quranic literature that elaborates on it. Instances of this narrative include Q Al-Hajj: 22–23:

> "Indeed, Allah will admit those who believe and do righteous deeds to gardens beneath which rivers flow. They will be adorned therein with bracelets of gold and pearl, and their garments therein will be silk";

and Q Al-Insan 76: 4:

> "We have prepared chains, iron collars, and blazing fire for the disbelievers".

Whatever the epistemic perspective adopted (literary, historical, philological, or hermeneutical), the scholarship frequently comments on the eschatological verses and texts by employing terms such as "symbolic", "metaphor", "poetization", "fictionalization", and other related rhetorical representational conceptualities. For example, in his literary analysis of the garden theme in the pre-Islamic poetry and Quranic paradisal imagery, Jaakko Hameen-Anttila generically calls this theme "the garden metaphor" (Hameen-Anttila 2017, pp. 136–61). In the Qur'an, however, this characterization becomes questionable when put to the test of the question that Ludwig Wittgenstein asks concerning the use of language to assert a fact or an idea: "but how is the connection between the name and the thing named set up?"[25] Does the connection that Hameen-Anttila sets up between his choice of the word "metaphor" and the Quranic paradisiac garden he thus names correspond to the connection that the Qur'an itself sets up between the language it employs to depict this theme and that which it intends to signify with it? By inference, putting aside the complex history of the Arabian literary tradition and instead focusing on the Quranic intentionality, ontotheology, and phenomenology of the sacred, we may wonder: does the garden imagery metaphorize an unimaginable eternal bliss or an intangible ideal of the afterlife, or does it describe an Islamic sacred paradise that would then have some tangible qualities linking it to the earthly domain through an ontotheological consubstantiality? Sebastian Günther 's critical assertions in his essay, "The Poetics of Islamic Eschatology", raise the same interrogation as he writes:

> "Analysing the wealth of images and symbols, the highly poetic language, and the complex web of arguments, all embedded in the often remarkably refined narrative structures of Arabic eschatological texts, may help us to better understand the ways in which descriptions of the next world, understood both (and sometimes simultaneously) as literal *and* figurative references to the hereafter, are instrumental for Muslim authors in communicating, vivifying, and reinforcing fundamental articles of Islamic faith." (Günther 2017, p. 182)

Further ahead in his essay, Günther argues "But even more than these powerful rhetorical or narrative features that address the mind and the emotions, it is the richness of symbolic imagery, metaphors, and colors so distinctive to these eschatological texts that effectively facilitates two fundamental objectives of the Quranic message as understood by generations of Muslims: the one is *missionary in its nature*, as non-Muslims are called upon to understand that acceptance of the Quranic message and Islam means salvation and eternal life; and the other is *dogmatic in its essence*, as it provides Muslims with reaffirmation of, and instruction in, Islamic doctrine" (Ibid, p. 210).

Although this critique by no means undermines these studies' value, in my view they typify two common correlated issues that are at stake in the problem of the Qur'an's use of symbolization: a certain mode of reasoning and the symbolic characterization itself. A hint at these issues is given by the labels of "poetics" and "the garden metaphor" that overall declare the eschatological verses and para-Quranic literature based on them as "highly symbolic" rhetorical poetic constructs. The first problem with this approach is that it overemphasizes the instrumentalization of the themes of paradise and hell for rhetorical and ideation-based didactic ends. In his essay, Günther indeed argues so insistently that the texts in question "reinforce", "facilitate", and are "instrumental for" missionary and dogmatic purposes that he overplays the Quranic eschatological narrative's didactic pragmatism at the expense of its primal endeavor, *revealing* the Islamic sacred factuality that is otherwise inaccessible to the experimental consciousness. To recall it, this sacred factuality consists of the resurrectional structure of Muslim existence. Thus, by underemphasizing the Qur'an's foundational act of revealing an unfathomable truth and foregrounding, by contrast, "the dual mission of reassuring Muslim believers and calling upon non-Muslims to convert to Islam", Günther rationalizes both the texts' content and the intent underpinning them (Ibid, pp. 190–91). And through this process of rationalization of the religious message, he effectively relegates at a second plane the latter's fundamental raison d'être, which is to expose the sacred data configuring the Islamic metaphysics centered on the afterlife.

The second issue is the loose use of the critical terminology of the metaphor and similar symbolizing linguistic concepts that not only support but also reinforce this rationalizing approach. This reinforcement necessarily takes place because employing this terminology recurrently without localizing and explaining its precise workings within the texts necessarily entails an overall localization of these texts in the ontological region of the symbolic. Following this way of presenting things, indeed, symbolic poetry covers with its creative veil the notion of the hereafter inconceivable to reason, which thereby becomes a conceivable construct of imagination. The modalities of prehension and penetration of this imaginary symbolic construct, of what lies behind it, and of what it truly signifies, are then left up to the Quranic addressees to sort out according to their disposition. To cite another instance of this reading, Günther describes in terms of "fictionalized presentations of the hereafter" the various para-Quranic elaborations seeking to explain the Qur'an's eschatological accounts by all manner of argument. A close examination heeding metaphor theory will show that, somehow assimilating Islamic eschatology to a poetic imaginary oeuvre akin to Dante's *Divine Comedy*, this reading does not stand up to the analysis of the texts' phenomenology of the sacred.

In *The Rules of Metaphor*, the philosopher Paul Ricoeur explains that "as figure, metaphor constitutes a displacement and an extension of the meaning of words. Its explanation is grounded in a theory of substitution" (Ricoeur 2003, p. 1). These terms of "figure", "extension", "displacement", and "substitution" signal the inscription of metaphor in representation as opposed to presentation, which is the work of the direct description. Pertaining to the interpretive language engaging the symbolizing imagination, the metaphor represents through an image but does not directly describe its object, be it a thing, being, or idea. This means that the metaphorical proposition projects a reshaped reality of this object, which is then not given in suchness. By contradistinction, the direct description is characterized by an unmediated closeness with its object, whose constitutive reality it aims to expose as truthfully as possible. In short, while metaphor represents, description presents.[26] This conceptual distinction is essential for identifying and explaining the presence of symbolic devices in a text, as opposed to straightforward descriptive accounts, and a fortiori for understanding this text's intentionality and semantic articulation.

In view of both this theoretical clarification and the Wittgensteinian differentiation between the act of naming and its semantic setup, the symbolizing interpretation at best maintains a bothersome ambiguity regarding the nature of the Islamic hereafter's imagery. While the Qur'an itself does not allow for attributing with certitude an ontological status to this imagery as either a mediating poetic fiction expressing an abstract notion of the afterlife

or a direct picture hinting at what it may consist of, this interpretation remains unclear about which semantic evidence it relies on. It does not tell us whether the symbols and metaphors that it talks about include the extensive depictions of gardens, flowing rivers, fire, etc. In other words, this interpretation does not consider carefully enough "the context of the situation" within the verses that Muhammad Abdel Haleem posits as an essential parameter for building a plausible Quranic hermeneutics (Abdel Haleem 2001, p. 160).[27] Yet, the scrutiny of this context of the situation does facilitate the detection—or, on the contrary, the invalidation—of a logic of substitution or representation in the eschatological narratives, beginning with the statements against poetry indicative of the Quranic intentionality, such as in Q, Ya-Sin, 36: 69:

> "And we did not give Prophet Muhammad the knowledge of poetry, nor is it befitting for him. It is not but a message and a clear Qur'an."[28]

## 6. The Quranic Realism

To continue with *Roads to Paradise*, Günther and Lawson aptly remark that in the Quranic imagery of paradise and hell "the spiritual is connected to the material", although, as just discussed, it seems that they locate this connection in the linguistic expression and leave open or vague, depending on how one sees it, what that statement entails for this imagery's ontological definition in the context of the eschatological situation (Günther and Lawson 2017, p. 18). At this level of the text's meaning, the two scholars prefer to rely on the Muslim theologians who, they stipulate, variably apprehend that connection between this and the next world at a literal or figurative level of meaning, or both simultaneously. But, again, the binary semantic model of the literal versus the figurative appears to be applied all too quickly to the Quranic eschatological accounts. For the text's two levels of signification implied by this model can be easily confused with the regular one-level mode of signifying which, however, may signify something that possesses a double ontology. This complex point requires some theoretical explanation.

The difference between these two semantic configurations resides in the crucial distinction between the signifying modality and the object itself that this modality intends to signify. The binary literal–figurative model of signification implicates the intervention of a representation (i.e., the figurative) of the signified entity in the signifying process, whereas the one-track semantic scheme does not. Whether this scheme is employed to discuss an entity constituted by a single or a double ontology has no bearing on the fact that the discussion itself operates on the direct non-representational mode, because the dualism, if there is one, resides in this entity—not in the discursive process. In both sacred scriptures and literature, the opposite to this scheme—the binary literal–figurative model—defines a whole range of representational semantic constructs combining fiction and reality, including tales with a historical substratum such as mythologies, parables, and allegories. The goal of these constructs is to rhetorically signify the real part through elements of fiction. For example, the myth of Sisyphus represents or illustrates a certain condition of human life by means of a fictive story. While Sisyphus is an imaginary representational figure, the narration in which this figure acts constitutes an imagistic fiction purposely invented to reveal the reality of said condition of human life. Unlike this mythological story of Sisyphus, the narration of, say, Jesus's death and resurrection in Christian scriptures does not rest upon such a double system of meaning involving elements of fiction because, according to the Christian creed, it directly and faithfully narrates the unfolding of a real event. This narration is therefore no mythology—not even a tale with a historical substratum from the Christian viewpoint—even though it revolves around a protagonist bestowed with a double ontotheology (human and divine) and an event that is both natural and supranatural.

In view of this mise au point with clear examples, the thorny question of the metaphor in the Qur'an resurfaces with more acuity, but this time in a differently formulated manner: does the Quranic eschatological narrative present the condition of meaning enabling us to define it as an Abrahamic monotheistic mythology or an allegory of the Islamic afterlife? Or does the context of the Quranic situation indicate that this narrative constitutes a sacred

"realistic" account in which the colorful depictions form an indexical (i.e., signaletic albeit asymbolic) semiosis about the afterlife as it will occur in due time?[29] Leaving aside the exegetical philosophy of the *zahir* and the *batin* applied by some Muslim thinkers to the entire Qur'an, the latter's textual situation does not provide any evidence that would permit a reading of the eschatological verses at a second representational degree beyond their literality, nor do these verses' apparent semantic structure. With these verses showing no obvious logic of substitution or mediation, we are consequently most likely dealing with the formula of a text univocally signifying something with a double ontology instead of the model of a doubly signifying text. The literal–figurative binary being thus ruled out, the thesis of the eschatological verses univocally describing an object constituted of a double ontology may be argued as follows:

Except for certain subcurrents of thought supporting a symbolic reading of these verses, Islamic theology traditionally conceives of them as the presentation of a sacred factuality. This state of affairs led Abdel Haleem to caution against the popular reductivist view of the "physical picture" of paradise as a locus dedicated to sensual pleasures and bodily wellbeing (Abdel Haleem 2001, p. 96). To him, although the Quranic afterlife is by no means "something theoretical", this view eclipses, overlooks, or undervalues this physical picture's spiritual dimension and significance, which he reinstates by placing the excerpts concerned in "a proper perspective" (Ibid, pp. 90, 97). To summarize Haleem's argument, in accordance with the Islamic principle of resurrection and, more broadly, with what I would describe as the corporeal anchorage of spirituality in the Qur'an, the attractive heavenly abode will frame an eternal life of spiritual order in the presence of God. Consequently, considering both the eschatological verses' semantic configuration and the dominant Muslim asymbolic comprehension of them, it can be said that the Quranic hereafter has a double ontology owing to its compound of material substance and spiritual essence—a double feature that the verses signify in all the directness of a descriptive presentation. It then appears logical that these two-level ontological complexities had been the object of a variety of theological interrogations and reflections on this matter of substance or essence, or on both aspects of the afterlife. For example, while some commentators have been particularly concerned with the difference of nature between the earthly and resurrected corporeity, others have placed their focus on the question of the appearance of God's face to the faithful in Paradise.

Thus, by applying the schema of the literal–figurative semantic device as a multipurpose critical tool, the scholarship often mistakes this Muslim theological pondering of the hereafter's double material–spiritual ontology for evidence of the double-signifying structure of the texts analyzed. But there is another reason for this mistake. While the Qur'an does posit corporeity and spirituality as the two indivisible components (i.e., substance and essence) of the one and the same sacred reality of the afterlife, and not as two autonomous ontologies—one concrete and symbolic (the materialist eschatological imagery) and one abstract and conceptual (the pure notion of eternal life)—linked by a representational or substitutive relationship, elsewhere in the suras it also frequently utilizes the binary schema in question. This fact undoubtedly creates confusion. An example of verse that does signify figuratively beyond its literality, Q, al-Naml, 27:44, will make the point clearer:

"She was told, 'Enter the palace.' But when she saw it, she thought it was a body of water and uncovered her shins. He said, 'Indeed, it is a palace made smooth with glass.' She said, 'My Lord, indeed I have wronged myself, and I submit with Solomon to Allāh, Lord of the worlds'."

This scenography of the encounter between Bilqis and Solomon in his glass palace works semantically as follows: The idolatrous queen's deceiving experience of the architecture demonstrates, in concrete visualizing terms, or metaphorizes both the triumph of Islam over paganism and the abstract thought that, although most severely misinformed, any well-disposed individual may acquire awareness of the true religion by God's will (see again (Gonzalez 2001, 2002)). In conformity with this semantic formula, the event described is not the cause or foundation of these religious ideals but a manifestation of

them (among other similar manifestations) in the Qur'an, which in this particular context of the Solomonic situation is evidenced by Bilqis's conversion. In phenomenological terms, these religious abstractions constitute "free idealities" independent of the Solomonic narrative medium that manifests or metaphorizes them, this very process of manifestation or metaphorization thus attesting to this medium's representational function by means of the literal–figurative binary.

To go further into the demonstration that what is viewed by scholars as metaphor in fact proceeds from the direct presentation, other elements signal that the connection between the material and the spiritual in the Islamic scripture noted by Günther and Lawson does not only concern the text's level of language (i.e., the name)—it also concerns its level of meaning (i.e., the semantic setup), where it signifies the existence of a "real" consubstantiality between this and the next world, "the visible world being a trace of that other world", as Al-Ghazali writes in *Mishkat al-Anwar*.[30] That the afterlife will be lived in a new creation made of a new materiality, which is yet to be discovered fully only at the advent of "the hour", does not contradict this assessment, because two things may be consubstantial without being ontologically identical. Thus, founding what I call "the Quranic realism"—i.e., the sacred made real through an explicit description of the human life's divine program within the Islamic cosmogonic structure—the consubstantiality between the two worlds is pointed to throughout the text. This pointing process can be identified through the Qur'an's uniquely insistent encouragement to practice analogy following the propositional model of "this is the sign, *ayat*, *isharat* or *mithal*, of that thing or situation", particularly through the frequent summation to engage the addressee in an observational attitude toward nature's marvels, presented as evidential forces of the intertwinement of the earthly and heavenly domains. Actually, by substituting "the Photograph" for the Quranic eschatological imagery, we may say with Roland Barthes that "nothing can prevent [this imagery =] the Photograph from being analogical" (Barthes 2000, p. 88). As the contemporary Muslim writer Mostafa Badawi explains, "He [God] provided them [the addressees] with ample indications, seen, heard, or otherwise perceived through the senses, to make them grasp that which lies beyond their senses... Everything in this world points to the next... We should always make the connection between what we perceive with our senses and what we cannot perceive but know exists" (Badawi 2020, pp. 85, 89). To support his propos, Badawi cites a hadith by the Companion Abu Razin al-'Uqayli who asked about seeing God in the afterlife:

> "Oh Messenger of God, will each of us see His Lord on Resurrection Day? And what is the sign of that in His creation? The Prophet asked him, 'Do you not gaze at the moon, all together, as if you were alone with it?'" (Al-Hakim, *Mustadrak* (8832); Ahmad, *Musnad* (15603), cited by Badawi 2020, p. 85).

But that is not all. More unusual features imbue the Qur'an with the realism of a living evidence, including a relational mode of speech delivery transforming the entire text into a kind of ongoing dialogue between the divine and its addressees. The philological studies have well deciphered the dynamic Quranic voicing system by means of which God engages and talks to its audience, alternating pronouns and multitoned configurations of questions, interjections, summations, and straightforward statements. Thanks to this versatile mode of communication, the divine speech does not just instruct and tell stories in a didactic authoritative manner. It interrogates, provokes, and harangues, thereby extracting reactive responses from the addressees, as in Q, Yunus, 31:

> "Say, 'Who provides for you from the heaven and the earth? Or who controls hearing and sight and who brings the living out of the dead and brings the dead out of the living and who arranges [every] matter?' They will say, 'Allah', so say, 'Then will you not fear Him?'

Called "participation" in 20th-century aesthetic theory and contemporary criticism on immersive art, these reactive responses expected from this relational mode of engagement help create proximity and intimacy between the author/artwork and the beholders.

Participation is therefore central to the texture of the Qur'an as an injunction to a total conscious–affective spiritual activity, as opposed to a mere distanced obedience to God's instructions ([Bishop 2006](#), p. 96). Crucially, this unprecedented participatory texture of the Qur'an not only reinforces its realism, but it also shapes it into a "situationist construct", to appropriate another concept of participation theory. Operating in the concreteness of the here and now, the situationist Qur'an breaks away with the conventional "once-upon a time" formulae of storytelling characterizing the Bible and New Testament. In effect, the holy book of Islam exits sacred history as we know it.

## 7. Participation, Immersion, Situationist Constructs: The Quranic "Theater of Cruelty"

In the famous manifesto *Toward a Situationist International* that he wrote in 1957, the artist and Marxist philosopher Guy-Ernest Debord explained the participation-based principle of the situationist construction:

> "Our central purpose is the construction of situations, that is, the concrete construction of temporary settings of life and their transformation into a higher, passionate nature. We must develop an intervention directed by the complicated factors of two great components in perpetual interaction: the material setting of life and the behaviour that it incites".[31]

In my view, this citation perfectly describes one of the fundamental Quranic intents if we replace the pronoun "our" with "God" and put the term "life" in the sacred perspective of the double Muslim existence in temporality and eternity. Of particular importance is "the material setting of life" necessary to fashion any situationist framework. In the context of religious scripture, it demands the suspension of the once-upon-a-time unfolding of history to bring about the here-and-now's spatiotemporality. As seen with its dialogic structure, which by definition temporarily takes the text out of history, in the Qur'an such suspensions of time occur regularly enough to collapse the past–present–future evolution into a perpetual present that will only end on the Day of Judgement. Until then, the situationist Qur'an allows/will allow every living soul to experience the divine speech not as a historical scripture transmitted through generations since its emergence in Arabia, but literally as "a happening" in the artistic sense, i.e., an immersive event witnessed and lived by the addressees who become participants at each reading, recitation, or hearing of this speech. For example, in Q, al-Mulk, 67: 2–4, the beholders find themselves facing the perfect geometric arrangement of the seven heavens as God challenges them to discern any flaw, as [if] they [were] are standing before it and staring at it:

> "He who created death and life to test you [as to] which of you is best in deed— and He is the Exalted in Might, the Forgiving–
>
> And who created seven heavens in layers. You do not see in the creation of the Most Merciful any inconsistency. So return your vision to the sky; do you see any breaks?
>
> Then return your vision twice again. Your vision will return to you humbled while it is fatigued."

Again, the scholarly mind seeking objectivity and the unbeliever may interpret these verses as a symbolic masterpiece of rhetoric conveying the abstract ideal of God's kingdom in the unseen beyond. As plausible as it may appear, this common interpretation nevertheless remains *objectively* flawed because it passes over two Quranic phenomenological features that are determining for the Quranic production of meaning as a religious scripture:

First, this interpretation amounts to ignoring the consistent effort of the Qur'an's author to project, throughout the text, a tangible cosmographic sketching of the Islamic cosmogony as a sacred reality—not a poetic fiction—and, as such, a supra-reality that cannot constitute itself in the human mind without the divine explanatory intervention.

Second, and most importantly, this interpretation does not heed the very particular cognitive system that the Quranic realistic texture induces precisely as a situationist opus pulling in the beholder's own life space in its lifelike spatiotemporal setup. It more

precisely concerns a cognitive system of eidetic order allowing for a perfect coordination between sensory–psychic and noetic–speculative processes of reception—although, due to the textual nature of the material, it goes without saying that this mode of empirical prehension of its discourse desired by the Qur'an cannot be bodily in the organic sense of the term. That mode can only rely on the eidetic phenomenology of sensation, whereby the senses' activation takes place in the mind, thanks to consciousness's ability to reconstitute and reimagine sensory processes proceeding from the experiential memory, that "vast structure of recollection" in Marcel Proust's words from *In Search of Lost time* (Proust 2023). The famous story of the madeleine in Proust's masterpiece indeed magnificently illustrates this extraordinary power of the experiential memory that the Qur'an exploits with maximal efficacy:

"The sight of the little madeleine had recalled nothing to my mind before I tasted it; perhaps because I had so often seen such things in the interval, without tasting them, on the trays in pastry-cooks' windows, that their image had dissociated itself from those Combray days to take its place among others more recent to take its place among others more recent; perhaps because of those memories, so long abandoned and put out of mind, nothing now survived, everything was scattered; the forms of things, including that of the little scallop-shell of pastry, so richly sensual under its severe, religious folds, were either obliterated or had been so long dormant as to have lost the power of expansion which would have allowed them to resume their place in my consciousness. But when from a long-distant past nothing subsists, after the people are dead, after the things are broken and scattered, still, alone, more fragile, but with more vitality, more unsubstantial, more persistent, more faithful, the smell and taste of things remain poised a long time, like souls, ready to remind us, waiting and hoping for their moment, amid the ruins of all the rest; and bear unfaltering, in the tiny and almost impalpable drop of their essence, the vast structure of recollection."[32]

Thus, like fine edible confectionaries, verbal artworks, including the divine artwork of the Qur'an, enable phenomena of mental recreation and re-enaction of things and situations that are not physically there or no longer there—in particular, the mental recreation of the sensory reality of the physiologically felt. In *Don Quixote*, Miguel de Cervantes gives a superb account of this vastly expansive eidetic phenomenology that, in the sphere of text, blows up any frontier between body and mind. About a poem sung that, without hesitation, could be replaced by Quranic verses recited, Cervantes writes:

"Those verses are no sooner heard, they presently produce a dancing of soul, tickling of fancies, emotions of spirit, and, in short, a pleasing distemper of the whole body, as if quicksilver shook it in every part." (de Cervantes Saavedra 1993, p. 169)

This citation also underscores the arousalist forces of the very moment or instant in which things happen, which emanate from the wording "no sooner heard" and "presently produce". Key to the aesthetic workings of a specific type of situationist construct in multimedia, texts, theater, and material artworks such as installations, these surging forces triggered by the very moment in which the event advents provoke participation in a particular manner that we may describe as "brutal", as they literally assault the senses and psyche without allowing the mind the time to react. This act of assaulting and its corporeal effects appear, for example, explicitly signified and eidetically stimulated in the vocabulary of Q, al-Mulk, 67: 4, previously quoted, as Muhammad Husain Kazi explains:

"The term "*karratayn*", de-rived from the root word "*karr*", is often used to describe someone who retreats ثُمَّ ٱرْجِعِ ٱلْبَصَرَ كَرَّتَيْنِ يَنقَلِبْ إِلَيْكَ ٱلْبَصَرُ خَاسِئًا وَهُوَ حَسِيرٌ ۝ from a battle, only to charge and lay assault once again. In the Quranic context, this word paints a lively metaphorical image of the addressee laying assault on the creation of al-Rahman, retreating, and then returning once again upon God's command. The fair outcome of these continuous assaults depends on the human faculty of *basar*, namely the cognitive faculty that allows one to lucidly

see the reality of things. After attempting to challenge the truth of this reality, the addressee will necessarily coil back, perplexed, chidden and driven away like an animal, as the word "*khasi'*" implies by connotation. The defeated gazer will be extremely fatigued and exhausted like a camel; an animal capable of traveling for days in the desert, but that slowly loses its energy during the long journey, eventually ending up unable to move."[33]

Thus, arousalism is central to the Qur'an's strategies of persuasion not only through the sublime poetic language's beauty, but also through the assault of the senses aiming to reach out to subjectivity's deepest folds as a path to divine knowledge. Contributing to its construction as a situationist holy scripture in which God and its addressees are engaged in a direct confrontation, this type of brutal arousalist technique lends the Qur'an's eidetic phenomenology of sensation a character of "violence" in the sense of a hyper-intensive form of physical and, a fortiori, mental tantalizing. Such a feature cannot but recall Antonin Artaud's "Theater of Cruelty", a major source of inspiration of the situationist movement represented by Debord, with which the Qur'an shares principles and techniques for producing meaning beyond the delivery of discourses and plots. Although the concept of cruelty in its expansive metaphysical sense embraces the idea of bodily pain, Artaud's definition of his conception of theater strikingly resonates with the Quranic intentionality as I comprehend it. Expressing his revolt against traditional theater in his landmark writing, *The Theater and its Double*, Artaud wrote: "We cannot go on prostituting the idea of theater whose only value is in its excruciating, magical relation to reality and danger" (Artaud 1958, p. 89). In the Quranic context, this reality is God's creation here and beyond, and the danger is the eternal life of suffering in hell to which the unbelievers condemn themselves. Only cruelty—the violent arousalism that will "flow into the sensibility" through that assaulting eidetic phenomenology of sensation, be it pain or pleasure—can help the divine message "to recover the notion of a kind of unique language half-way between gesture and thought" (Ibid. pp. 89 and 91). See how, to warn of the danger of hell, the Qur'an unleashes intense sensory forces on the body that the audience may eidetically feel in anticipation of what awaits them in the afterlife if they persist in unbelieving, in Qur'an, at-Tawbah, 9: 35:

> "The Day when it will be heated in the fire of Hell and seared therewith will be their foreheads, their flanks, and their backs, 'This is what you hoarded for yourselves, so taste what you used to hoard'."

Finally, like in Artaud's creations, in "the Quranic theater of cruelty" the power of sound carries the whole thing and takes it to another level of physical–psychic intensity: "it turns words into incantations. It extends the voice. It utilizes the vibrations and qualities of the voice… It seeks to exalt, to benumb, to charm, to arrest the sensibility. It liberates a new lyricism of gesture which, by its precipitation or its amplitude in the air, ends by surpassing the lyricism of words. It ultimately breaks away from the intellectual subjugation of the language, by conveying the sense of a new and deeper intellectuality which hides itself beneath the gestures and signs, raised to the dignity of particular exorcisms" (Artaud 1958, p. 91).

## 8. Conclusions: The Qur'an as a Divine Screenplay of the Muslim Play of Life

Hopefully, the findings presented in this essay will convince readers that Islam is so much more than a discursive tradition. This is because its foundation, God's speech in the Qur'an, radically subverts discourse itself as the Judeo-Christian scriptures conceptualize it following the age-old model of historical unfolding from which emerge teachings and doctrinal principles. Although it maintains the historical layout, the Qur'an integrates it into a situation of dialogue, as a constant reminder. But this dialogue is no mere conversation. It is an eventful encounter perpetually reenacted like a play at each moment of the text's reception, during which God submits the audience to all sort of tests and trials and shakes their subjectivities to the core by calling up the awakening and revelatory forces of sensation and affect. It thus makes participation the motor of its divine screenplay of the Muslim play of life. Echoing the definition by Western theorists of certain global aesthetic practices

as "art of participation", the arousalist Qur'an makes of Islam "a religion of participation". In this hermeneutic light, it should then appear clearly that processes of symbolization play a much less important role in the Quranic theater of cruelty than one generally thinks. Symbols and metaphors punctuate its screenplay but do not shape it, as its situationist structure and texture demonstrate that the Qur'an is no monotheist mythology. It is instead the concreteness of a sacred factuality divinely revealed in the directness of pain and pleasure and made felt eidetically that does. And how could we have come to such conclusions? By taking another radically open critical path—transcultural, interdisciplinary, and centered on the phenomenological method scrutinizing unchartered spaces of meaning-making and consciousness. Notably, or at least I hope, the phenomenology of the sacred, as a rational method of inquiry, has proven to be efficient in the tackling of the challenges that historians face in understanding holy scriptures, as Fred Donner limpidly expounded in his Presidential Address at the 2022 International Quranic Studies Association meeting in Palermo. I believe, indeed, that this method offers a way to reconcile the traditional rational–historical approach with the religious viewpoint. For, by no means involving or reflecting a personal embracing of the faith, this type of phenomenological inquiry apprehends this viewpoint as a kind of evidence as objective and critically important as the text itself.

**Funding:** This research received no external funding.

**Institutional Review Board Statement:** Not applicable.

**Informed Consent Statement:** Not applicable.

**Data Availability Statement:** No new data were created or analyzed in this study. Data sharing is not applicable to this article.

**Acknowledgments:** I sincerely thank Susanne Olsson for her invitation to contribute to this Special Issue of Religions, 'Religions in 2022'. I am also grateful to Hannelies Koloska who organized a panel on her project, "Vision and Visuality in the Qur'an and Beyond", for the International Quranic Studies Association (IQSA) 2022 meeting, in Palermo, in which she kindly invited me to participate and during which I presented some of the ideas developed in this essay. I express my heartfelt thanks to Gabriel S. Reynolds who, after this meeting, encouraged me to write down these ideas. I finally thank also the three reviewers for their insights that have helped me to improve this article.

**Conflicts of Interest:** The author declares no conflict of interest.

## Notes

[1] Islamic sonic studies are emerging and eye-opening. About sound, although it is discorporate and ephemeral, it remains a material form belonging to the material world. See, for example (Eisenlohr 2018); this book offers a well-informed theoretical framework for this type of enquiry.

[2] See the discussions and the citation in (Asad 2009, p. 1).

[3] Many studies deal with this topic but Anne-Sylvie Boisliveau has dedicated an entire monograph on it (Boisliveau 2014), see my (online) review/essay of this book in (Gonzalez 2017).

[4] I discuss this concept of Islamic logocentrism in (Gonzalez 2021, pp. 6–33).

[5] The concept of affect has generated a great variety of theories and interpretations over time, from Baruch Spinoza to Gilles Deleuze, leading in the 21st century to the "affective turn" in the humanities at large. To summarize, affect has been conceptualized in two ways: as an elemental state provoked by an external stimulus of whatever nature—what Antonio Damasio describes as "the feeling of what happens" in (Damasio 1999); and as an intensive transformational force that produces different states or affections "at a material, presubjective, asignifying level", to quote Brian L. Ott in (Ott 2017, p. 10). This clear and concise entry on the theoretical history of affect offers a useful bibliography on the subject.

[6] For detailed studies on these varied somatic concepts see (Chittick 2013; Ingenito 2021; Zargar 2008; Puerta Vílchez 2017; Gonzalez 2001, 2014).

[7] To be noted, affect theory and phenomenology in general distinguish between sensation, feeling, and emotion (see Ott 2017; Altieri 2003; and Thrift 2008).

8　　Allusion to Richard Wagner's conception of *Gesamtkunstwerk*, "the total art work" that is comprehensive and all-embracing of all art forms, for a more profound and complete aesthetic experience. See https://interlude.hk/richard-wagners-concept-of-the-gesamtkunstwerk/ accessed on 14 June 2023.

9　　See "3-Expression and Meaning" in (Welton 1999, pp. 26–51) and (Benenti 2020).

10　　(Benenti 2020) is the reference to consult for a thorough theoretical presentation of the trope of "the expressive object".

11　　As examples of application of this threefold epistemology see (Gonzalez 2001, 2002, pp. 26–42), and (Gonzalez 2019, pp. 187–204). Phenomenology is not only a multifarious branch of philosophy, but also an analytical method for the study of subjects and objects in humanities at large, and in certain sciences such as neurosciences and cognitive psychology. For easy access to this complex material, see the entry "Phenomenology" in the very useful online *Stanford Encyclopedia of Philosophy*: (Smith 2018), and again (Welton 1999).

12　　The distinction is to be made between the Quranic text as an artwork and the material art that it has inspired throughout history. About this material art, see (Suleman 2007).

13　　See my discussion on the problem of religion in Islamic art display in (Gonzalez 2022).

14　　Within this broad configuration, there are also, on the one hand, Muslim scholars representing the countercurrent who also work in the West, in parallel with the secularist mainstream; and on the other hand, experts with a Muslim heritage who, having been educated in the West, do follow this mainstream. However, I did not come across a secularist scholarship produced by institutions in the Muslim-majority countries.

15　　"Modern" here signifies the period spanning from modernism in the late 19th century to contemporaneity. Regarding the use of this material, some rare scholars in Islamic textual studies do engage in it to advance their work. See, for example, (Almond 2004; Melvin-Koushki 2016).

16　　See "1.6 Arousalism: Matravers, Ridley and Robinson on Art and Emotions", in (Benenti 2020, pp. 31–38). It should be noted that, while the term "perceptive" is related to perception in general, which engages altogether the senses, the psyche and the mind, the term "perceptual" is related only to the corporeal–sensory function of perception, also called "perceptuality".

17　　A concise definition of rasa and reading references can be found in the online *Encyclopaedia Britannica* (2023).

18　　Just to be clear, this discussion only concerns the canonical rituals in which religious representatives utter the holy book—not the singing rites that are performed outside these rituals, which in the case of Islam are not consensually accepted. The Sufi sonic practices are therefore excluded from this discussion.

19　　About the Quranic recitation in the context of its emergence, see (Boisliveau and Hilali 2022).

20　　This observation heeds the fact that not all listeners, Muslim and non-Muslim, have the necessary education and command of the Arabic language to understand fully the Quranic semantics.

21　　Although this is an old academic work, it remains a classic reference for the extraordinarily rich painting-based aesthetic philosophy of ancient China.

22　　In *Roads to Paradise*, the introduction surveys the rich corpus of studies dealing with poetry, symbols, and metaphors in the Qur'an.

23　　All Quranic citations in this essay are taken from *Sahih International*. About the metaphors in poetry in primary sources, see (Abu Deeb 1979). See also the case study of a Quranic Solomonic parable in (Gonzalez 2002), and Chapter 2 "The Aesthetics of the Solomonic Parable in the Qur'an" in (Gonzalez 2001, pp. 26–41).

24　　I find particularly useful and resort frequently to paraphrasing writers and artists whose ideas and statements I deem appropriate and enlightening for the study of Islam. This technique, aptly called "ventriloquism" by the theorist of performing arts Camilla Damkjær, has been used by the philosopher Gilles Deleuze himself. In her non-published outstanding doctoral dissertation (Damkjær 2005), she explains "One of the methods that characterise Deleuze's work almost throughout his career, is his subtle way of miming his materials. Miming in the manner almost of a ventriloquist, making his materials speak, making for instance Henri Bergson, Baruch de Spinoza, Friedrich Nietzsche and others speak… But this ventriloquism has nothing to do with imitation or even reproduction. Deleuze makes them speak with his own voice, his own choice of perspectives, his own arguments".

25　　Wittgenstein asks this question in taking the example of the use of the term "pain" in Section 244 in (Wittgenstein 1968, p. 89).

26　　It goes without saying that these conceptual distinctions between representation and presentation must be understood within the broader framework of language that is, by nature, representational.

27　　Abdel Haleem notably cites Abu Ishaq al-Shatibi in his *Muwafaqat*, stating "The science of *ma'aní* [meaning] and *bayan* [factual and figurative expression] by which the *i'jaz* [inimitability] of the Qur'an is recognised, revolves around knowing the requirements of the situation during the discourse from the point of view of the discourse itself, the discursant, the discursee or all of them together; for the same statement can be understood in different ways in relation to two different addressees or more".

28　　This statement seems to contradict the fact that the Qur'an does resort to poetry in its rhetoric. But, again, the contradiction fades if we consider the difference between the linguistic tools and what they signify in relation to both intentionality and the "context of the situation".

29　　The term "indexical" draws from Peircean semiotics. See (Atkin 2023). In contrast with the two other types of signs in Charles S. Peirce's semiotic—namely, the symbol and the icon—the index points to something but does not represent it. The study of

the Quranic signaletic vocabulary, including *ayat*, *isharat*, and *mithal*, would greatly benefit from the application of this theory in a separate essay. I believe that the Qur'an presents a rich array of semantic configurations depending on the context of the situation in which these signaletic terms appear, although I intuit that the indexical configuration largely dominates, like in the eschatological narrative. The index presents the particular semantic property of keeping a certain ontological space between itself as a sign and what it pinpoints, which is important in the Quranic case that talks about ungraspable sacred otherworldly realities by putting them in relation to the living phenomenal realities—a future research perspective!

30   Al-Ghazali, *Mishkat al-Anwar*, 1:4, cited by (Badawi 2020, p. 91). See also, alongside Günther's cited study, Abdel Haleem, parts 7 and 8, "Life and Beyond", and "Paradise in the Quran", in (Abdel Haleem 2001, pp. 82–106).

31   For a good overview of the varied theories and aesthetic movements related to this concept, which had significant repercussions beyond Europe, in North and South America, see (Bishop 2006). See also the landmark text (Bourriaud 1998).

32   See the excerpt in (Proust 2023) *In Search of Lost time* Available online with more excerpts from the same text about sense-memory in 'Swann's Way': https://www.berfrois.com/2013/05/swanns-way-marcel-proust/?fbclid=IwAR2SkGhLK-iEf4DtsHJEXE_Mi6n9eY5mPGbTeadhMCHJK31fE2b0wqiGyeQ (accessed on 14 June 2023).

33   This is my own rephrased report of Muhammad Husain Kasi's philological study of this verse in (Gonzalez 2019, p. 199).

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
