# Peer review of "Phenomenology of Quranic Corporeality and Affect: A Concrete Sense of Being Muslim in the World"

_religions, doi:10.3390/rel14070827_

Round 1
Reviewer 1 Report
The author is in complete command of Qur'anic interpretation, and he/she operates comfortably within a complexly nuanced interdisciplinary approach. Moreover, the article is well researched and passionately argued. There are a few observations that I believe the paper shouldn't be published until they are addressed:
- The author rightfully critiques Orientalist readings of Islamic texts, including the Qur'an. This is necessary, encouraged, and applauded. The author simultaneously critiques what he/she calls "the secularist mainstream" scholars and Muslim scholars (both in and outside of the West) who embrace what he/she calls "the spirituality-based interpretation." Again, this is a welcome and much-needed balance whose absence plagues the field of Islamic studies everywhere (although the terms "secularist" [academic?] and "spirituality-based" [faith-based?] need reconsideration/clarification). However, the author may have not practiced what he/she preached. In certain parts of the article, he/she seems to be fully embracing the so-called "spirituality-based interpretation." For example, “Symbols and metaphors punctuate its screenplay but do not shape it, as its situationist structure and texture demonstrate that the Qur’an is no monotheist mythology. It is the concreteness of sacred facts divinely revealed in the directness of pain and pleasure made felt eidetically that does.” (p. 18)
- Awkwardly constructed, extremely long sentences that are very difficult to follow. Here's one of many examples:
“But again, the literal-figurative semantic binary appears to be applied to the eschatological accounts all too quickly, as it can be easily confused with a non-binary semantic configuration whereby the text signifies literally something that, however, possesses a double ontology” (p. 12)
Sporadic references to "humans" as "man" also punctuate the article (for example, “Man and the divine” Line 5). It is recommended that the author ask a colleague or an editor to review the language and style of the article to achieve a smoother, more gender-neutral articulations of these sentences, especially, in the abstract/introduction and conclusion, which, I believe, needs to be re-written. Finally, very few typos need to be weeded out (for example, “all sort of tests” (p. 18).
Author Response
I sincerely thank this reviewer for their appreciation and careful critical reading of my essay. I will rework it according to their concerns. I will replace 'spirituality-based' by 'faith-based' as suggested, but I believe that the term 'secularist is appropriate in my critique of the art historical mainstream. This scholarship uses the term 'secular' routinely. For example, in the Victoria and Albert Museum, in the rooms dedicated to Islamic art one may encounter the label "Secular Mamelouk Art".
About the point of contention: 'that the Qur'an is no monotheist mythology', I will clarify in my text this notion of mythology. I oppose it to 'sacred history and facts' within a religious system. There is a significant difference, in my view, between these notions as mythology is a 'representation' of religious thought through a narrative construct mixing fiction with elements of history, whereas sacred history gives itself directly as facts without any representational mediation; 'facts' which are sacred because they are unverifiable historically. Sacred facts naturally can be combined with historical facts in scriptures. In my article, my interpretive point is that certain Quranic excerpts perceived as symbolic by the scholarship, e.i. representational or mythological, instead expound sacred facts. This point is the result of an analytical examination of semantics, not from a spiritualist approach that would involve a personal spiritual engagement with the Qur'an.
Reviewer 2 Report
Footnotes 21, 22 empty -I assume they may quote the author's name: fix it if it's not the case-.
Author Response
I sincerely thank this reviewer for their very positive evaluation of my text.
Regarding the empty footnotes, their content have been indeed deliberately obliterated in order to insure anonymity as required.
Reviewer 3 Report
This a profound, original, and well-argued article. My only suggestion is that the language be simplified a little bit to make it accessible to a wider audience, especially undergraduates, who will probably be unable to process many of its ideas due to the technical jargon and sophisticated nature of the argument. There are a few typos that the author should also address. One on line 246 (Abu, should be Abu Musa). The second is on line 392 (use of preposition). The last concerns the word "although", which is divided at the wrong place.
Author Response
I am most grateful for this reviewer's appreciation of the article. I will certainly correct the mistakes they have indicated in their comment, and will review my writing style so as to make it more fluid and easier to read.
Round 2
Reviewer 1 Report
The author successfully incorporated the comments given on their original manuscript. The revised version is improved and acceptable.
One final proofreading of the article is recommended.